# STABLE RECURRENT MODELS

**John Miller & Moritz Hardt**
University of California, Berkeley
`{miller_john,hardt}@berkeley.edu`

## ABSTRACT

Stability is a fundamental property of dynamical systems, yet to this date it has had little bearing on the practice of recurrent neural networks. In this work, we conduct a thorough investigation of stable recurrent models. Theoretically, we prove stable recurrent neural networks are well approximated by feed-forward networks for the purpose of both inference and training by gradient descent. Empirically, we demonstrate stable recurrent models often perform as well as their unstable counterparts on benchmark sequence tasks. Taken together, these findings shed light on the effective power of recurrent networks and suggest much of sequence learning happens, or can be made to happen, in the stable regime. Moreover, our results help to explain why in many cases practitioners succeed in replacing recurrent models by feed-forward models.

## 1    INTRODUCTION

Recurrent neural networks are a popular modeling choice for solving sequence learning problems arising in domains such as speech recognition and natural language processing. At the outset, recurrent neural networks are non-linear dynamical systems commonly trained to fit sequence data via some variant of gradient descent.

Stability is of fundamental importance in the study of dynamical system. Surprisingly, however, stability has had little impact on the practice of recurrent neural networks. Recurrent models trained in practice do not satisfy stability in an obvious manner, suggesting that perhaps training happens in a chaotic regime. The difficulty of training recurrent models has compelled practitioners to successfully replace recurrent models with non-recurrent, feed-forward architectures.

This state of affairs raises important unresolved questions. *Is sequence modeling in practice inherently unstable? When and why are recurrent models really needed?*

In this work, we shed light on both of these questions through a theoretical and empirical investigation of stability in recurrent models.

We first prove stable recurrent models can be approximated by feed-forward networks. In particular, not only are the models equivalent for *inference*, they are also equivalent for *training* via gradient descent. While it is easy to contrive non-linear recurrent models that on some input sequence cannot be approximated by feed-forward models, our result implies such models are inevitably unstable. This means in particular they must have exploding gradients, which is in general an impediment to learnibility via gradient descent.

Second, across a variety of different sequence tasks, we show how *recurrent models can often be made stable without loss in performance*. We also show models that are nominally unstable often operate in the stable regime on the data distribution. Combined with our first result, these observation helps to explain why an increasingly large body of empirical research succeeds in replacing recurrent models with feed-forward models in important applications, including translation (Vaswani et al., 2017; Gehring et al., 2017), speech synthesis (Van Den Oord et al., 2016), and language modeling (Dauphin et al., 2017). While stability does not always hold in practice to begin with, it is often possible to generate a high-performing stable model by *imposing stability* during training.

Our results also shed light on the effective representational properties of recurrent networks trained in practice. In particular, stable models cannot have long-term memory. Therefore, when stable and

unstable models achieve similar results, either the task does not require long-term memory, or the unstable model does not have it.

## 1.1 CONTRIBUTIONS

In this work, we make the following contributions.

1. We present a generic definition of stable recurrent models in terms of non-linear dynamical systems and show how to ensure stability of several commonly used models. Previous work establishes stability for vanilla recurrent neural networks. We give new sufficient conditions for stability of long short-term memory (LSTM) networks. These sufficient conditions come with an efficient projection operator that can be used at training time to enforce stability.

2. We prove, under the stability assumption, feed-forward networks can approximate recurrent networks for purposes of both inference and training by gradient descent. While simple in the case of inference, the training result relies on non-trivial stability properties of gradient descent.

3. We conduct extensive experimentation on a variety of sequence benchmarks, show stable models often have comparable performance with their unstable counterparts, and discuss when, if ever, there is an intrinsic performance price to using stable models.

## 2 STABLE RECURRENT MODELS

In this section, we define *stable recurrent models* and illustrate the concept for various popular model classes. From a pragmatic perspective, stability roughly corresponds to the criterion that the gradients of the training objective do not *explode* over time. Common recurrent models can operate in both the stable and unstable regimes, depending on their parameters. To study stable variants of common architectures, we give sufficient conditions to ensure stability and describe how to efficiently enforce these conditions during training.

## 2.1 DEFINING STABLE RECURRENT MODELS

A *recurrent model* is a non-linear dynamical system given by a differentiable *state-transition map* $\phi_w \colon \mathbf{R}^n \times \mathbf{R}^d \to \mathbf{R}^n$, parameterized by $w \in \mathbf{R}^m$. The hidden state $h_t \in \mathbf{R}^n$ evolves in discrete time steps according to the update rule

$$h_t = \phi_w(h_{t-1}, x_t)\,, \tag{1}$$

where the vector $x_t \in \mathbf{R}^d$ is an arbitrary input provided to the system at time $t$. This general formulation allows us to unify many examples of interest. For instance, for a recurrent neural network, given weight matrices $W$ and $U$, the state evolves according to

$$h_t = \phi_{W,U}(h_{t-1}, x_t) = \tanh\left(W h_{t-1} + U x_t\right).$$

Recurrent models are typically trained using some variant of gradient descent. One natural—even if not strictly necessary—requirement for gradient descent to work is that the gradients of the training objective do not explode over time. *Stable recurrent models* are precisely the class of models where the gradients cannot explode. They thus constitute a natural class of models where gradient descent can be expected to work. In general, we define a stable recurrent model as follows.

**Definition 1.** *A recurrent model $\phi_w$ is stable if there exists some $\lambda < 1$ such that, for any weights $w \in \mathbf{R}^m$, states $h, h' \in \mathbf{R}^n$, and input $x \in \mathbf{R}^d$,*

$$\|\phi_w(h, x) - \phi_w(h', x)\| \leq \lambda \|h - h'\|\,. \tag{2}$$

Equivalently, a recurrent model is stable if the map $\phi_w$ is $\lambda$-contractive in $h$. If $\phi_w$ is $\lambda$-stable, then $\|\nabla_h \phi_w(h, x)\| < \lambda$, and for Lipschitz loss $p$, $\|\nabla_w p\|$ is always bounded (Pascanu et al., 2013).

Stable models are particularly well-behaved and well-justified from a theoretical perspective. For instance, at present, only *stable* linear dynamical systems are known to be learnable via gradient

descent (Hardt et al., 2018). In unstable models, the gradients of the objective can explode, and it is a delicate matter to even show that gradient descent converges to a stationary point. The following proposition offers one such example. The proof is provided in the appendix.

**Proposition 1.** *There exists an unstable system $\phi_w$ where gradient descent does not converge to a stationary point, and $\|\nabla_w p\| \to \infty$ as the number of iterations $N \to \infty$.*

## 2.2 Examples of Stable Recurrent Models

In this section, we provide sufficient conditions to ensure stability for several common recurrent models. These conditions offer a way to require learning happens in the stable regime– after each iteration of gradient descent, one imposes the corresponding stability condition via projection.

**Linear dynamical systems and recurrent neural networks.** Given a Lipschitz, point-wise non-linearity $\rho$ and matrices $W \in \mathbf{R}^{n \times n}$ and $U \in \mathbf{R}^{n \times d}$, the state-transition map for a recurrent neural network (RNN) is

$$h_t = \rho(Wh_{t-1} + Ux_t).$$

If $\rho$ is the identity, then the system is a linear dynamical system. Jin et al. (1994) show if $\rho$ is $L_\rho$-Lipschitz, then the model is stable provided $\|W\| < \frac{1}{L_\rho}$. Indeed, for any states $h, h'$, and any $x$,

$$\|\rho(Wh + Ux) - \rho(Wh' + Ux)\| \le L_\rho \|Wh + Ux - Wh' - Ux\| \le L_\rho \|W\| \|h - h'\|.$$

In the case of a linear dynamical system, the model is stable provided $\|W\| < 1$. Similarly, for the 1-Lipschitz $\tanh$-nonlinearity, stability obtains provided $\|W\| < 1$. In the appendix, we verify the assumptions required by the theorems given in the next section for this example. Imposing this condition during training corresponds to projecting onto the spectral norm ball.

**Long short-term memory networks.** Long Short-Term Memory (LSTM) networks are another commonly used class of sequence models (Hochreiter & Schmidhuber, 1997). The state is a pair of vectors $s = (c, h) \in \mathbf{R}^{2d}$, and the model is parameterized by eight matrices, $W_\square \in \mathbf{R}^{d \times d}$ and $U_\square \in \mathbf{R}^{d \times n}$, for $\square \in \{i, f, o, z\}$. The state-transition map $\phi_{\text{LSTM}}$ is given by

$$
\begin{aligned}
f_t &= \sigma(W_f h_{t-1} + U_f x_t) \\
i_t &= \sigma(W_i h_{t-1} + U_i x_t) \\
o_t &= \sigma(W_o h_{t-1} + U_o x_t) \\
z_t &= \tanh(W_z h_{t-1} + U_z x_t) \\
c_t &= i_t \circ z_t + f_t \circ c_{t-1} \\
h_t &= o_t \cdot \tanh(c_t),
\end{aligned}
$$

where $\circ$ denotes elementwise multiplication, and $\sigma$ is the logistic function.

We provide conditions under which the iterated system $\phi_{\text{LSTM}}^r = \phi_{\text{LSTM}} \circ \cdots \circ \phi_{\text{LSTM}}$ is stable. Let $\|f\|_\infty = \sup_t \|f_t\|_\infty$. If the weights $W_f, U_f$ and inputs $x_t$ are bounded, then $\|f\|_\infty < 1$ since $|\sigma| < 1$ for any finite input. This means the next state $c_t$ must "forget" a non-trivial portion of $c_{t-1}$. We leverage this phenomenon to give sufficient conditions for $\phi_{\text{LSTM}}$ to be contractive in the $\ell_\infty$ norm, which in turn implies the iterated system $\phi_{\text{LSTM}}^r$ is contractive in the $\ell_2$ norm for $r = O(\log(d))$. Let $\|W\|_\infty$ denote the induced $\ell_\infty$ matrix norm, which corresponds to the maximum absolute row sum $\max_i \sum_j |W_{ij}|$.

**Proposition 2.** *If $\|W_i\|_\infty, \|W_o\|_\infty < (1 - \|f\|_\infty)$, $\|W_z\|_\infty \le (1/4)(1 - \|f\|_\infty)$, $\|W_f\|_\infty < (1 - \|f\|_\infty)^2$, and $r = O(\log(d))$, then the iterated system $\phi_{\text{LSTM}}^r$ is stable.*

The proof is given in the appendix. The conditions given in Proposition 2 are fairly restrictive. Somewhat surprisingly we show in the experiments models satisfying these stability conditions still achieve good performance on a number of tasks. We leave it as an open problem to find different parameter regimes where the system is stable, as well as resolve whether the original system $\phi_{\text{LSTM}}$ is stable. Imposing these conditions during training and corresponds to simple row-wise normalization of the weight matrices and inputs. More details are provided in Section 4 and the appendix.

# 3 STABLE RECURRENT MODELS HAVE FEED-FORWARD APPROXIMATIONS

In this section, we prove stable recurrent models can be well-approximated by feed-forward networks for the purposes of both inference and training by gradient descent. From a memory perspective, stable recurrent models are *equivalent* to feed-forward networks—both models use the same amount of context to make predictions. This equivalence has important consequences for sequence modeling in practice. When a stable recurrent model achieves satisfactory performance on some task, a feed-forward network can achieve similar performance. Consequently, if sequence learning in practice is inherently stable, then recurrent models may not be necessary. Conversely, if feed-forward models cannot match the performance of recurrent models, then sequence learning in practice is in the unstable regime.

## 3.1 TRUNCATED RECURRENT MODELS

For our purposes, the salient distinction between a recurrent and feed-forward model is the latter has *finite-context*. Therefore, we say a model is *feed-forward* if the prediction made by the model at step $t$ is a function only of the inputs $x_{t-k}, \ldots, x_t$ for some finite $k$.

While there are many choices for a feed-forward approximation, we consider the simplest one— truncation of the system to some finite context $k$. In other words, the feed-forward approximation moves over the input sequence with a sliding window of length $k$ producing an output every time the sliding window advances by one step. Formally, for context length $k$ chosen in advance, we define the *truncated model* via the update rule

$$h_t^k = \phi_w(h_{t-1}^k, x_t), \quad h_{t-k}^k = 0 \,. \tag{3}$$

Note that $h_t^k$ is a function only of the previous $k$ inputs $x_{t-k}, \ldots, x_t$. While this definition is perhaps an abuse of the term "feed-forward", the truncated model can be implemented as a standard autoregressive, depth-$k$ feed-forward network, albeit with significant weight sharing.

Let $f$ denote a prediction function that maps a state $h_t$ to outputs $f(h_t) = y_t$. Let $y_t^k$ denote the predictions from the truncated model. To simplify the presentation, the prediction function $f$ is not parameterized. This is without loss of generality because it is always possible to fold the parameters into the system $\phi_w$ itself. In the sequel, we study $\left\| y_t - y_t^k \right\|$ both during and after training.

## 3.2 APPROXIMATION DURING INFERENCE

Suppose we train a full recurrent model $\phi_w$ and obtain a prediction $y_t$. For an appropriate choice of context $k$, the truncated model makes essentially the same prediction $y_t^k$ as the full recurrent model. To show this result, we first control the difference between the hidden states of both models.

**Lemma 1.** *Assume $\phi_w$ is $\lambda$-contractive in $h$ and $L_x$-Lipschitz in $x$. Assume the input sequence $\|x_t\| \le B_x$ for all $t$. If the truncation length $k \ge \log_{1/\lambda}\left(\frac{L_x B_x}{(1-\lambda)\varepsilon}\right)$, then the difference in hidden states $\left\| h_t - h_t^k \right\| \le \varepsilon$.*

Lemma 1 effectively says stable models do not have long-term memory– distant inputs do not change the states of the system. A proof is given in the appendix. If the prediction function is Lipschitz, Lemma 1 immediately implies the recurrent and truncated model make nearly identical predictions.

**Proposition 3.** *If $\phi_w$ is a $L_x$-Lipschitz and $\lambda$-contractive map, and $f$ is $L_f$ Lipschitz, and the truncation length $k \ge \log_{1/\lambda}\left(\frac{L_f L_x B_x}{(1-\lambda)\varepsilon}\right)$, then $\left\| y_t - y_t^k \right\| \le \varepsilon$.*

## 3.3 APPROXIMATION DURING TRAINING VIA GRADIENT DESCENT

Equipped with our inference result, we turn towards optimization. We show gradient descent for stable recurrent models finds essentially the same solutions as gradient descent for truncated models. Consequently, both the recurrent and truncated models found by gradient descent make essentially the same predictions.

Our proof technique is to initialize both the recurrent and truncated models at the same point and track the divergence in weights throughout the course of gradient descent. Roughly, we show if

$k \approx O(\log(N/\varepsilon))$, then after $N$ steps of gradient descent, the difference in the weights between the recurrent and truncated models is at most $\varepsilon$. Even if the gradients are similar for both models at the same point, it is a priori possible that slight differences in the gradients accumulate over time and lead to divergent weights where no meaningful comparison is possible. Building on similar techniques as Hardt et al. (2016), we show that gradient descent itself is stable, and this type of divergence cannot occur.

Our gradient descent result requires two essential lemmas. The first bounds the difference in gradient between the full and the truncated model. The second establishes the gradient map of both the full and truncated models is Lipschitz. We defer proofs of both lemmas to the appendix.

Let $p_T$ denote the loss function evaluated on recurrent model after $T$ time steps, and define $p_T^k$ similarly for the truncated model. Assume there some compact, convex domain $\Theta \subset \mathbf{R}^m$ so that the map $\phi_w$ is stable for all choices of parameters $w \in \Theta$.

**Lemma 2.** *Assume $p$ (and therefore $p^k$) is Lipschitz and smooth. Assume $\phi_w$ is smooth, $\lambda$-contractive, and Lipschitz in $x$ and $w$. Assume the inputs satisfy $\|x_t\| \leq B_x$, then*

$$\left\| \nabla_w p_T - \nabla_w p_T^k \right\| = \gamma k \lambda^k,$$

*where $\gamma = O\left(B_x(1-\lambda)^{-2}\right)$, suppressing dependence on the Lipschitz and smoothness parameters.*

**Lemma 3.** *For any $w, w' \in \Theta$, suppose $\phi_w$ is smooth, $\lambda$-contractive, and Lipschitz in $w$. If $p$ is Lipschitz and smooth, then*

$$\|\nabla_w p_T(w) - \nabla_w p_T(w')\| \leq \beta \|w - w'\|,$$

*where $\beta = O\left((1-\lambda)^{-3}\right)$, suppressing dependence on the Lipschitz and smoothness parameters.*

Let $w_{\text{recurr}}^i$ be the weights of the recurrent model on step $i$ and define $w_{\text{trunc}}^i$ similarly for the truncated model. At initialization, $w_{\text{recurr}}^0 = w_{\text{trunc}}^0$. For $k$ sufficiently large, Lemma 2 guarantees the difference between the gradient of the recurrent and truncated models is negligible. Therefore, after a gradient update, $\left\| w_{\text{recurr}}^1 - w_{\text{trunc}}^1 \right\|$ is small. Lemma 3 then guarantees that this small difference in weights does not lead to large differences in the gradient on the subsequent time step. For an appropriate choice of learning rate, formalizing this argument leads to the following proposition.

**Proposition 4.** *Under the assumptions of Lemmas 2 and 3, for compact, convex $\Theta$, after $N$ steps of projected gradient descent with step size $\alpha_t = \alpha/t$, $\left\| w_{\text{recurr}}^N - w_{\text{trunc}}^N \right\| \leq \alpha \gamma k \lambda^k N^{\alpha\beta+1}$.*

The decaying step size in our theorem is consistent with the regime in which gradient descent is known to be stable for non-convex training objectives (Hardt et al., 2016). While the decay is faster than many learning rates encountered in practice, classical results nonetheless show that with this learning rate gradient descent still converges to a stationary point; see p. 119 in Bertsekas (1999) and references there. In the appendix, we give empirical evidence the $O(1/t)$ rate is necessary for our theorem and show examples of stable systems trained with constant or $O(1/\sqrt{t})$ rates that do not satisfy our bound.

Critically, the bound in Proposition 4 goes to 0 as $k \to \infty$. In particular, if we take $\alpha = 1$ and $k \geq \Omega(\log(\gamma N^\beta/\varepsilon))$, then after $N$ steps of projected gradient descent, $\left\| w_{\text{recurr}}^N - w_{\text{trunc}}^N \right\| \leq \varepsilon$. For this choice of $k$, we obtain the main theorem. The proof is left to the appendix.

**Theorem 1.** *Let $p$ be Lipschitz and smooth. Assume $\phi_w$ is smooth, $\lambda$-contractive, Lipschitz in $x$ and $w$. Assume the inputs are bounded, and the prediction function $f$ is $L_f$-Lipschitz. If $k \geq \Omega(\log(\gamma N^\beta/\varepsilon))$, then after $N$ steps of projected gradient descent with step size $\alpha_t = 1/t$, $\left\| y_T - y_T^k \right\| \leq \varepsilon$.*

## 4 EXPERIMENTS

In the experiments, we show stable recurrent models can achieve solid performance on several benchmark sequence tasks. Namely, we show unstable recurrent models can often be made stable without a loss in performance. In some cases, there is a small gap between the performance between unstable and stable models. We analyze whether this gap is indicative of a "price of stability" and show the unstable models involved are stable in a data-dependent sense.

### 4.1 TASKS

We consider four benchmark sequence problems–word-level language modeling, character-level language modeling, polyphonic music modeling, and slot-filling.

**Language modeling.** In language modeling, given a sequence of words or characters, the model must predict the next word or character. For character-level language modeling, we train and evaluate models on Penn Treebank (Marcus et al., 1993). To increase the coverage of our experiments, we train and evaluate the word-level language models on the Wikitext-2 dataset, which is twice as large as Penn Treebank and features a larger vocabulary (Merity et al., 2017). Performance is reported using bits-per-character for character-level models and perplexity for word-level models.

**Polyphonic music modeling.** In polyphonic music modeling, a piece is represented as a sequence of 88-bit binary codes corresponding to the 88 keys on a piano, with a 1 indicating a key that is pressed at a given time. Given a sequence of codes, the task is to predict the next code. We evaluate our models on JSB Chorales, a polyphonic music dataset consisting of 382 harmonized chorales by J.S. Bach (Allan & Williams, 2005). Performance is measured using negative log-likelihood.

**Slot-filling.** In slot filling, the model takes as input a query like "I want to Boston on Monday" and outputs a class label for each word in the input, e.g. Boston maps to Departure_City and Monday maps to Departure_Time. We use the Airline Travel Information Systems (ATIS) benchmark and report the F1 score for each model (Price, 1990).

### 4.2 COMPARING STABLE AND UNSTABLE MODELS

For each task, we first train an unconstrained RNN and an unconstrained LSTM. All the hyperparameters are chosen via grid-search to maximize the performance of the unconstrained model. For consistency with our theoretical results in Section 3 and stability conditions in Section 2.2, both models have a single recurrent layer and are trained using plain SGD. In each case, the resulting model is unstable. However, we then retrain the best models using projected gradient descent to enforce stability *without retuning the hyperparameters*. In the RNN case, we constrain $\|W\| < 1$. After each gradient update, we project the $W$ onto the spectral norm ball by computing the SVD and thresholding the singular values to lie in $[0, 1)$. In the LSTM case, after each gradient update, we normalize each row of the weight matrices to satisfy the sufficient conditions for stability given in Section 2.2. Further details are given in the appendix.

**Stable and unstable models achieve similar performance.** Table 1 gives a comparison of the performance between stable and unstable RNNs and LSTMs on each of the different tasks. Each of the reported metrics is computed on the held-out test set. We also show a representative comparison of learning curves for word-level language modeling and polyphonic music modeling in Figures 1(a) and 1(b).

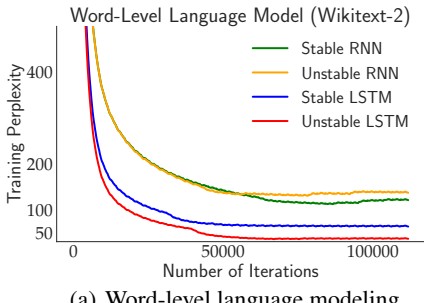
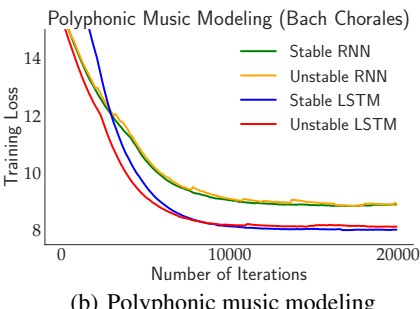

(a) Word-level language modeling          (b) Polyphonic music modeling

Figure 1: Stable and unstable variants of common recurrent architectures achieve similar performance across a range of different sequence tasks.

Table 1: Comparison of stable and unstable models on a variety of sequence modeling tasks. For all the tasks, stable and unstable RNNs achieve the same performance. For polyphonic music and slot-filling, stable and unstable LSTMs achieve the same results. On language modeling, there is a small gap between stable and unstable LSTMs. We discuss this in Section 4.3. Performance is evaluated on the held-out test set. For negative log-likelihood (nll), bits per character (bpc), and perplexity, lower is better. For F1 score, higher is better.

| | | Model | | | |
| | | RNN | | LSTM | |
| **Sequence Task** | **Dataset** (measure) | Unstable | Stable | Unstable | Stable |
|---|---|---|---|---|---|
| Polyphonic Music | JSB Chorales (nll) | 8.9 | 8.9 | 8.5 | 8.5 |
| Slot-Filling | Atis (F1 score) | 94.7 | 94.7 | 95.1 | 94.6 |
| Word-level LM | Wikitext-2 (perplexity) | 146.7 | 143.5 | 95.7 | 113.2 |
| Character-level LM | Penn Treebank (bpc) | 1.8 | 1.9 | 1.4 | 1.9 |

Across all the tasks we considered, stable and unstable RNNs have roughly the same performance. Stable RNNs and LSTMs achieve results comparable to published baselines on slot-filling (Mesnil et al., 2015) and polyphonic music modeling (Bai et al., 2018). On word and character level language modeling, both stable and unstable RNNs achieve comparable results to (Bai et al., 2018).

On the language modeling tasks, however, there is a gap between stable and unstable LSTM models. Given the restrictive conditions we place on the LSTM to ensure stability, it is surprising they work as well as they do. Weaker conditions ensuring stability of the LSTM could reduce this gap. It is also possible imposing stability comes at a cost in representational capacity required for some tasks.

### 4.3 WHAT IS THE "PRICE OF STABILITY" IN SEQUENCE MODELING?

The gap between stable and unstable LSTMs on language modeling raises the question of whether there is an intrinsic performance cost for using stable models on some tasks. If we measure stability in a data-dependent fashion, then the unstable LSTM language models are stable, indicating this gap is illusory. However, in some cases with short sequences, instability can offer modeling benefits.

**LSTM language models are stable in a "data-dependent" way.** Our notion of stability is conservative and requires stability to hold for every input and pair of hidden states. If we instead consider a weaker, data-dependent notion of stability, the word and character-level LSTM models are stable (in the iterated sense of Proposition 2). In particular, we compute the stability parameter only *using input sequences from the data*. Furthermore, we only evaluate stability on hidden states *reachable via gradient descent*. More precisely, to estimate $\lambda$, we run gradient ascent to find worst-case hidden states $h, h'$ to maximize $\frac{\left\| \phi_w(h,x) - \phi_w(h',x) \right\|}{\|h - h'\|}$. More details are provided in the appendix.

The data-dependent definition given above is a useful diagnostic— when the sufficient stability conditions fail to hold, the data-dependent condition addresses whether the model is still operating in the stable regime. Moreover, when the input representation is fixed during training, our theoretical results go through without modification when using the data-dependent definition.

Using the data-dependent measure, in Figure 2(a), we show the iterated character-level LSTM, $\phi_{\text{LSTM}}^r$, is stable for $r \approx 80$ iterations. A similar result holds for the word-level language model for $r \approx 100$. These findings are consistent with experiments in Laurent & von Brecht (2017) which find LSTM trajectories converge after approximately 70 steps *only when evaluated on sequences from the data*. For language models, the "price of stability" is therefore much smaller than the gap in Table 1 suggests– even the "unstable" models are operating in the stable regime on the data distribution.

**Unstable systems can offer performance improvements for short-time horizons.** When sequences are short, training unstable models is less difficult because exploding gradients are less of an issue. In these case, unstable models can offer performance gains. To demonstrate this, we train truncated unstable models on the polyphonic music task for various values of the truncation parameter $k$. In Figure 2(b), we simultaneously plot the performance of the unstable model and the stability

parameter $\lambda$ for the converged model for each $k$. For short-sequences, the final model is more unstable ($\lambda \approx 3.5$) and achieves a better test-likelihood. For longer sequence lengths, $\lambda$ decreases closer to the stable regime ($\lambda \approx 1.5$), and this improved test-likelihood performance disappears.

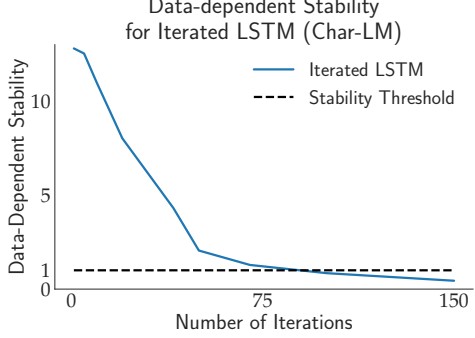

(a) Data-dependent stability of character-level language models. The iterated-LSTM refers to the iteration system $\phi^r_{\text{LSTM}} = \phi_{\text{LSTM}} \circ \cdots \circ \phi_{\text{LSTM}}$.

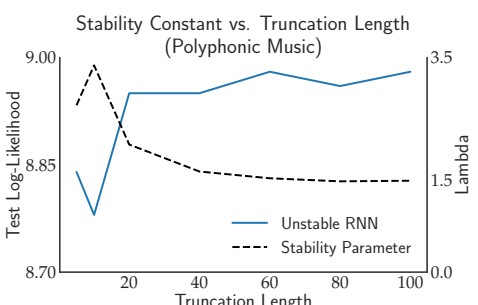

(b) Unstable models can boost performance for short sequences.

Figure 2: What is the intrinsic "price of stability"? For language modeling, we show the unstable LSTMs are actually stable in weaker, data-dependent sense. On the other hand, for polyphonic music modeling with short sequences, instability can improve model performance.

## 4.4 Unstable Models Operate in the Stable Regime

In the previous section, we showed nominally unstable models often satisfy a data-dependent notion of stability. In this section, we offer further evidence unstable models are operating in the stable regime. These results further help explain why stable and unstable models perform comparably in experiments.

**Vanishing gradients.** Stable models necessarily have vanishing gradients, and indeed this ingredient is a key ingredient in the proof of our training-time approximation result. For both word and character-level language models, we find both *unstable RNNs and LSTMs also exhibit vanishing gradients*. In Figures 3(a) and 3(b), we plot the average gradient of the loss at time $t + i$ with respect to the input at time $t$, $\|\nabla_{x_t} p_{t+i}\|$ as $t$ ranges over the training set. For either language modeling task, the LSTM and the RNN suffer from limited sensitivity to distant inputs at initialization and throughout training. The gradients of the LSTM vanish more slowly than those of the RNN, but both models exhibit the same qualitative behavior.

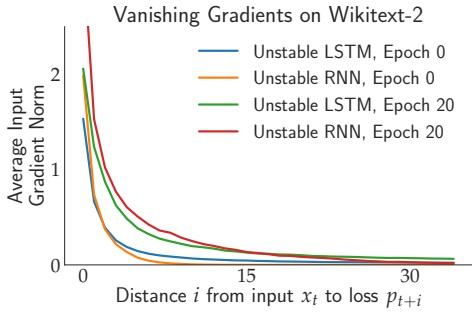

(a) Word-Level language modeling

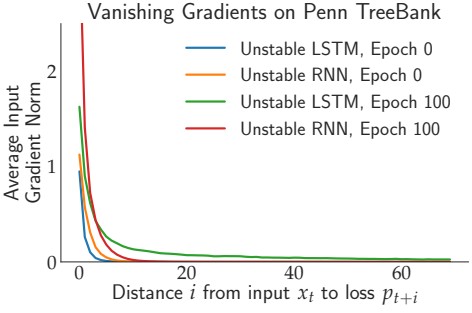

(b) Character-level language modeling

Figure 3: Unstable word and character-level language models exhibit vanishing gradients. We plot the norm of the gradient with respect to inputs, $\|\nabla_{x_t} p_{t+i}\|$, as the distance between the input and the loss grows, averaged over the entire training set. The gradient vanishes for moderate values of $i$ for both RNNs and LSTMs, though the decay is slower for LSTMs.

**Truncating Unstable Models.** The results in Section 3 show stable models can be truncated without loss of performance. In practice, unstable models can also be truncated without performance loss. In Figures 4(a) and 4(b), we show the performance of both LSTMs and RNNs for various values of the truncation parameter $k$ on word-level language modeling and polyphonic music modeling. Initially, increasing $k$ increases performance because the model can use more context to make predictions. However, in both cases, there is diminishing returns to larger values of the truncation parameter $k$. LSTMs are unaffected by longer truncation lengths, whereas the performance of RNNs slightly degrades as $k$ becomes very large, possibly due to training instability. In either case, diminishing returns to performance for large values of $k$ means truncation and therefore feed-forward approximation is possible even for these unstable models.

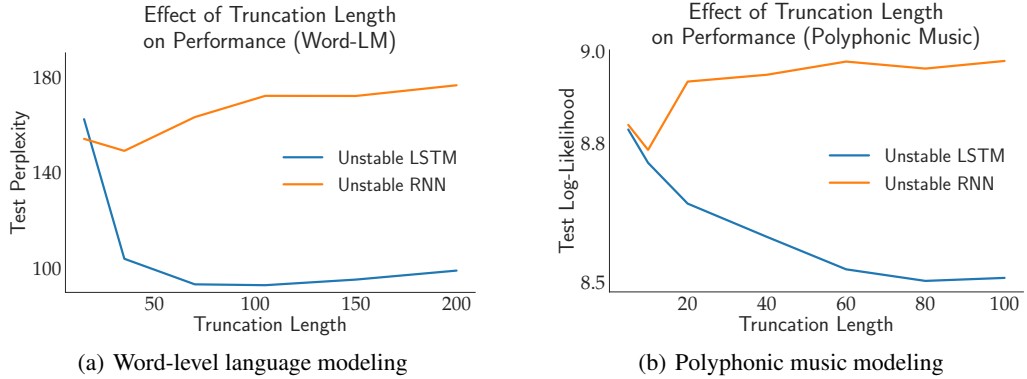

(a) Word-level language modeling  (b) Polyphonic music modeling

Figure 4: Effect of truncating unstable models. On both language and music modeling, RNNs and LSTMs exhibit diminishing returns for large values of the truncation parameter $k$. In LSTMs, larger $k$ doesn't affect performance, whereas for unstable RNNs, large $k$ slightly decreases performance

**Proposition (4) holds for unstable models.** In stable models, Proposition (4) in Section 3 ensures the distance between the weight matrices $\|w_{\mathrm{recurr}} - w_{\mathrm{trunc}}\|$ grows slowly as training progresses, and this rate decreases as $k$ becomes large. In Figures 5(a) and 5(b), we show a similar result holds empirically for unstable word-level language models. All the models are initialized at the same point, and we track the distance between the hidden-to-hidden matrices $W$ as training progresses. Training the full recurrent model is impractical, and we assume $k = 65$ well captures the full-recurrent model. In Figures 5(a) and 5(b), we plot $\|W_k - W_{65}\|$ for $k \in \{5, 10, 15, 25, 35, 50, 64\}$ throughout training. As suggested by Proposition (4), after an initial rapid increase in distance, $\|W_k - W_{65}\|$ grows slowly, as suggested by Proposition 4. Moreover, there is a diminishing return to choosing larger values of the truncation parameter $k$ in terms of the accuracy of the approximation.

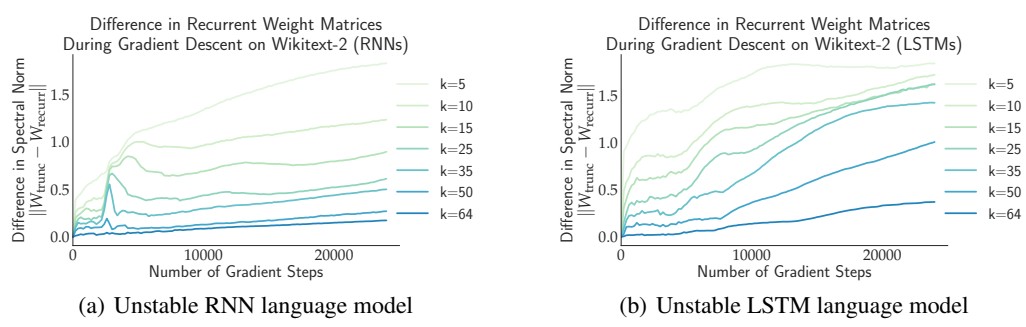

(a) Unstable RNN language model  (b) Unstable LSTM language model

Figure 5: Qualitative version of Proposition 4 for unstable, word-level language models. We assume $k = 65$ well-captures the full-recurrent model and plot $\|w_{\mathrm{trunc}} - w_{\mathrm{recurr}}\| = \|W_k - W_{65}\|$ as training proceeds, where $W$ denotes the recurrent weights. As Proposition 4 suggests, this quantity grows slowly as training proceeds, and the rate of growth decreases as $k$ increases.

## 5 ARE RECURRENT MODELS TRULY NECESSARY?

Our experiments show recurrent models trained in practice operate in the stable regime, and our theoretical results show stable recurrent models are approximable by feed-forward networks, As a consequence, we conjecture *recurrent networks trained in practice are always approximable by feed-forward networks*. Even with this conjecture, we cannot yet conclude recurrent models as commonly conceived are unnecessary. First, our present proof techniques rely on truncated versions of recurrent models, and truncated recurrent architectures like LSTMs may provide useful inductive bias on some problems. Moreover, implementing the truncated approximation as a feed-forward network increases the number of weights by a factor of $k$ over the original recurrent model. Declaring recurrent models truly superfluous would require both finding more parsimonious feed-forward approximations and proving natural feed-forward models, e.g. fully connected networks or CNNs, can approximate stable recurrent models during training. This remains an important question for future work.

## 6 RELATED WORK

Learning dynamical systems with gradient descent has been a recent topic of interest in the machine learning community. Hardt et al. (2018) show gradient descent can efficiently learn a class of stable, linear dynamical systems, Oymak (2018) shows gradient descent learns a class of stable, non-linear dynamical systems. Work by Sedghi & Anandkumar (2016) gives a moment-based approach for learning some classes of stable non-linear recurrent neural networks. Our work explores the theoretical and empirical consequences of the stability assumption made in these works. In particular, our empirical results show models trained in practice can be made closer to those currently being analyzed theoretically without large performance penalties.

For *linear* dynamical systems, Tu et al. (2017) exploit the connection between stability and truncation to learn a truncated approximation to the full stable system. Their approximation result is the same as our inference result for linear dynamical systems, and we extend this result to the non-linear setting. We also analyze the impact of truncation on training with gradient descent. Our training time analysis builds on the stability analysis of gradient descent in Hardt et al. (2016), but interestingly uses it for an entirely different purpose. Results of this kind are completely new to our knowledge.

For RNNs, the link between vanishing and exploding gradients and $\|W\|$ was identified in Pascanu et al. (2013). For 1-layer RNNs, Jin et al. (1994) give sufficient conditions for stability in terms of the norm $\|W\|$ and the Lipschitz constant of the non-linearity. Our work additionally considers LSTMs and provides new sufficient conditions for stability. Moreover, we study the consequences of stability in terms of feed-forward approximation.

A number of recent works have sought to avoid vanishing and exploding gradients by ensuring the system is an isometry, i.e. $\lambda = 1$. In the RNN case, this amounts to constraining $\|W\| = 1$ (Arjovsky et al., 2016; Wisdom et al., 2016; Jing et al., 2017; Mhammedi et al., 2017; Jose et al., 2018). Vorontsov et al. (2017) observes strictly requiring $\|W\| = 1$ reduces performance on several tasks, and instead proposes maintaining $\|W\| \in [1 - \varepsilon, 1 + \varepsilon]$. Zhang et al. (2018) maintains this "soft-isometry" constraint using a parameterization based on the SVD that obviates the need for the projection step used in our stable-RNN experiments. Kusupati et al. (2018) sidestep these issues and stabilizes training using a residual parameterization of the model. At present, these unitary models have not yet seen widespread use, and our work shows much of the sequence learning in practice, even with nominally unstable models, actually occurs in the stable regime.

From an empirical perspective, Laurent & von Brecht (2017) introduce a non-chaotic recurrent architecture and demonstrate it can perform as well more complex models like LSTMs. Bai et al. (2018) conduct a detailed evaluation of recurrent and convolutional, feed-forward models on a variety of sequence modeling tasks. In diverse settings, they find feed-forward models outperform their recurrent counterparts. Their experiments are complimentary to ours; we find recurrent models can often be replaced with stable recurrent models, which we show are equivalent to feed-forward networks.

ACKNOWLEDGEMENTS

This material is based upon work supported by the National Science Foundation Graduate Research Fellowship Program under Grant No. DGE 1752814 and a generous grant from the AWS Cloud Credits for Research program.

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

# A    Proofs from Section 2

## A.1    Gradient descent on unstable systems need not converge

*Proof of Proposition 1.*   Consider a scalar linear dynamical system

$$h_t = ah_{t-1} + bx_t \tag{4}$$
$$\hat{y}_t = h_t, \tag{5}$$

where $h_0 = 0$, $a, b \in \mathbf{R}$ are parameters, and $x_t, y_t \in \mathbf{R}$ are elements the input-output sequence $\{(x_t, y_t)\}_{t=1}^T$, where $L$ is the sequence length, and $\hat{y}_t$ is the prediction at time $t$. Stability of the above system corresponds to $|a| < 1$.

Suppose $(x_t, y_t) = (1, 1)$ for $t = 1, \ldots, L$. Then the desired system (4) simply computes the identity mapping. Suppose we use the squared-loss $\ell(y_t, \hat{y}_t) = (1/2)(y_t - \hat{y}_t)^2$, and suppose further $b = 1$, so the problem reduces to learning $a = 0$. We first compute the gradient. Compactly write

$$h_t = \sum_{i=0}^{t-1} a^t b = \left( \frac{1 - a^t}{1 - a} \right).$$

Let $\delta_t = (\hat{y}_t - y_t)$. The gradient for step $T$ is then

$$\frac{d}{da}\ell(y_T, \hat{y}_T) = \delta_T \frac{d}{da} = \delta_T \sum_{t=0}^{T-1} a^{T-1-t} h_t$$

$$= \delta_T \sum_{t=0}^{T-1} a^{T-1-t} \left( \frac{1 - a^t}{1 - a} \right)$$

$$= \delta_T \left[ \frac{1}{(1-a)} \sum_{t=0}^{T-1} a^t - \frac{Ta^{T-1}}{(1-a)} \right]$$

$$= \delta_T \left[ \frac{(1 - a^T)}{(1-a)^2} - \frac{Ta^{T-1}}{(1-a)} \right].$$

Plugging in $y_t = 1$, this becomes

$$\frac{d}{da}\ell(y_T, \hat{y}_T) = \left( \frac{(1 - a^T)}{(1-a)} - 1 \right) \left[ \frac{(1 - a^T)}{(1-a)^2} - \frac{Ta^{T-1}}{(1-a)} \right]. \tag{6}$$

For large $T$, if $|a| > 1$, then $a^L$ grows exponentially with $T$ and the gradient is approximately

$$\frac{d}{da}\ell(y_T, \hat{y}_T) \approx \left( a^{T-1} - 1 \right) Ta^{T-2} \approx Ta^{2T-3}$$

Therefore, if $a^0$ is initialized outside of $[-1, 1]$, the iterates $a^i$ from gradient descent with step size $\alpha_i = (1/i)$ diverge, i.e. $a^i \to \infty$, and from equation (6), it is clear that such $a^i$ are not stationary points.                                                                                                                                          □

## A.2    Proofs from section 2.2

### A.2.1    Recurrent neural networks

Assume $\|W\| \le \lambda < 1$ and $\|U\| \le B_U$. Notice $\tanh'(x) = 1 - \tanh(x)^2$, so since $\tanh(x) \in [-1, 1]$, $\tanh(x)$ is 1-Lipschitz and 2-smooth. We previously showed the system is stable since, for any states $h, h'$,

$$\|\tanh(Wh + Ux) - \tanh(Wh' + Ux)\|$$
$$\le \|Wh + Ux - Wh' - Ux\|$$
$$\le \|W\| \|h - h'\|.$$

Using Lemma 1 with $k = 0$, $\|h_t\| \leq \frac{B_U B_x}{(1-\lambda)}$ for all $t$. Therefore, for any $W, W', U, U'$,

$$\|\tanh(Wh_t + Ux) - \tanh(W'h_t + U'x)\|$$
$$\leq \|Wh_t + Ux - W'h_t - U'x\|$$
$$\leq \sup_t \|h_t\| \|W - W'\| + B_x \|U - U'\|.$$
$$\leq \frac{B_U B_x}{(1-\lambda)} \|W - W'\| + B_x \|U - U'\|,$$

so the model is Lipschitz in $U, W$. We can similarly argue the model is $B_U$ Lipschitz in $x$. For smoothness, the partial derivative with respect to $h$ is

$$\frac{\partial \phi_w(h, x)}{\partial h} = \mathbf{diag}(\tanh'(Wh + Ux))W,$$

so for any $h, h'$, bounding the $\ell_\infty$ norm with the $\ell_2$ norm,

$$\left\| \frac{\partial \phi_w(h, x)}{\partial h} - \frac{\partial \phi_w(h', x)}{\partial h} \right\| = \left\| \mathbf{diag}(\tanh'(Wh + Ux))W - \mathbf{diag}(\tanh'(Wh' + Ux))W \right\|$$
$$\leq \|W\| \left\| \mathbf{diag}(\tanh'(Wh + Ux) - \tanh'(Wh' + Ux)) \right\|$$
$$\leq 2 \|W\| \|Wh + Ux - Wh' - Ux\|_\infty$$
$$\leq 2\lambda^2 \|h - h'\|.$$

For any $W, W', U, U'$ satisfying our assumptions,

$$\left\| \frac{\partial \phi_w(h, x)}{\partial h} - \frac{\partial \phi_{w'}(h, x)}{\partial h} \right\| = \left\| \mathbf{diag}(\tanh'(Wh + Ux))W - \mathbf{diag}(\tanh'(W'h + U'x))W' \right\|$$
$$\leq \left\| \mathbf{diag}(\tanh'(Wh + Ux) - \tanh'(W'h + U'x)) \right\| \|W\|$$
$$\quad + \left\| \mathbf{diag}(\tanh'(W'h + U'x)) \right\| \|W - W'\|$$
$$\leq 2\lambda \|(W - W')h + (U - U')x\|_\infty + \|W - W'\|$$
$$\leq 2\lambda \|(W - W')\| \|h\| + 2\lambda \|U - U'\| \|x\| + \|W - W'\|$$
$$\leq \frac{2\lambda B_U B_x + (1-\lambda)}{(1-\lambda)} \|W - W'\| + 2\lambda B_x \|U - U'\|.$$

Similar manipulations establish $\frac{\partial \phi_w(h,x)}{\partial w}$ is Lipschitz in $h$ and $w$.

### A.2.2 LSTMs

Similar to the previous sections, we assume $s_0 = 0$.

The state-transition map is not Lipschitz in $s$, much less stable, unless $\|c\|$ is bounded. However, assuming the weights are bounded, we first prove this is always the case.

**Lemma 4.** *Let $\|f\|_\infty = \sup_t \|f_t\|_\infty$. If $\|W_f\|_\infty < \infty$, $\|U_f\|_\infty < \infty$, and $\|x_t\|_\infty \leq B_x$, then $\|f\|_\infty < 1$ and $\|c_t\|_\infty \leq \frac{1}{(1-\|f\|_\infty)}$ for all $t$.*

*Proof of Lemma 4.* Note $|\tanh(x)|, |\sigma(x)| \leq 1$ for all $x$. Therefore, for any $t$, $\|h_t\|_\infty = \|o_t \circ \tanh(c_t)\|_\infty \leq 1$. Since $\sigma(x) < 1$ for $x < \infty$ and $\sigma$ is monotonically increasing

$$\|f_t\|_\infty \leq \sigma\left(\|W_f h_{t-1} + U_f x_t\|_\infty\right)$$
$$\leq \sigma\left(\|W_f\|_\infty \|h_{t-1}\|_\infty + \|U_f\|_\infty \|x_t\|_\infty\right)$$
$$\leq \sigma\left(B_W + B_u B_x\right)$$
$$< 1.$$

Using the trivial bound, $\|i_t\|_\infty \leq 1$ and $\|z_t\|_\infty \leq 1$, so

$$\|c_{t+1}\|_\infty = \|i_t \circ z_t + f_t \circ c_t\|_\infty \leq 1 + \|f_t\|_\infty \|c_t\|_\infty.$$

Unrolling this recursion, we obtain a geometric series

$$\|c_{t+1}\|_\infty \leq \sum_{i=0}^{t} \|f_t\|_\infty^i \leq \frac{1}{(1 - \|f\|_\infty)}.$$

$\square$

*Proof of Proposition 2.* We show $\phi_{\text{LSTM}}$ is $\lambda$-contractive in the $\ell_\infty$-norm for some $\lambda < 1$. For $r \geq \log_{1/\lambda}(\sqrt{d})$, this in turn implies the iterated system $\phi_{\text{LSTM}}^r$ is contractive is the $\ell_2$-norm.

Consider the pair of reachable hidden states $s = (c, h)$, $s' = (c', h')$. By Lemma 4, $c, c'$ are bounded. Analogous to the recurrent network case above, since $\sigma$ is $(1/4)$-Lipschitz and $\tanh$ is 1-Lipschitz,

$$\|i - i'\| \leq \frac{1}{4} \|W_i\|_\infty \|h - h'\|_\infty$$

$$\|f - f'\| \leq \frac{1}{4} \|W_f\|_\infty \|h - h'\|_\infty$$

$$\|o - o'\| \leq \frac{1}{4} \|W_o\|_\infty \|h - h'\|_\infty$$

$$\|z - z'\| \leq \|W_z\|_\infty \|h - h'\|_\infty.$$

Both $\|z\|_\infty, \|i\|_\infty \leq 1$ since they're the output of a sigmoid. Letting $c_+$ and $c'_+$ denote the state on the next time step, applying the triangle inequality,

$$\begin{aligned}
\|c_+ - c'_+\|_\infty &\leq \|i \circ z - i' \circ z'\|_\infty + \|f \circ c - f' \circ c'\|_\infty \\
&\leq \|(i - i') \circ z\|_\infty + \|i' \circ (z - z')\|_\infty + \|f \circ (c - c')\|_\infty + \|c \circ (f - f')\|_\infty \\
&\leq \|i - i'\|_\infty \|z\|_\infty + \|z - z'\|_\infty \|i'\|_\infty + \|c - c'\|_\infty \|f\|_\infty + \|f - f'\|_\infty \|c\|_\infty \\
&\leq \left( \frac{\|W_i\|_\infty + \|c\|_\infty \|W_f\|_\infty}{4} + \|W_z\|_\infty \right) \|h - h'\|_\infty + \|f\|_\infty \|c - c'\|_\infty.
\end{aligned}$$

A similar argument shows

$$\left\| h_+ - h'_+ \right\|_\infty \leq \|o - o'\|_\infty + \left\| c_+ - c'_+ \right\|_\infty \leq \frac{\|W_o\|_\infty}{4} \|h - h'\|_\infty + \left\| c_+ - c'_+ \right\|_\infty.$$

By assumption,

$$\left( \frac{\|W_i\|_\infty + \|c\|_\infty \|W_f\|_\infty + \|W_o\|_\infty}{4} + \|W_z\|_\infty \right) < 1 - \|f\|_\infty,$$

and so

$$\left\| h_+ - h'_+ \right\|_\infty < (1 - \|f\|_\infty) \|h - h'\|_\infty + \|f\|_\infty \|c - c'\|_\infty \leq \|s - s'\|_\infty,$$

as well as

$$\left\| c_+ - c'_+ \right\|_\infty < (1 - \|f\|_\infty) \|h - h'\|_\infty + \|f\|_\infty \|c - c'\|_\infty \leq \|s - s'\|_\infty,$$

which together imply

$$\left\| s_+ - s'_+ \right\|_\infty < \|s - s'\|_\infty,$$

establishing $\phi_{\text{LSTM}}$ is contractive in the $\ell_\infty$ norm. $\square$

## B  PROOFS FROM SECTION 3

Throughout this section, we assume the initial state $h_0 = 0$. Without loss of generality, we also assume $\phi_w(0, 0) = 0$ for all $w$. Otherwise, we can reparameterize $\phi_w(h, x) \mapsto \phi_w(h, x) - \phi_w(0, 0)$ without affecting expressivity of $\phi_w$. For stable models, we also assume there some compact, convex domain $\Theta \subset \mathbf{R}^m$ so that the map $\phi_w$ is stable for all choices of parameters $w \in \Theta$.

*Proof of Lemma 1.* For any $t \geq 1$, by triangle inequality,

$$\|h_t\| = \|\phi_w(h_{t-1}, x_t) - \phi_w(0, 0)\| \leq \|\phi_w(h_{t-1}, x_t) - \phi_w(0, x_t)\| + \|\phi_w(0, x_t) - \phi_w(0, 0)\|.$$

Applying the stability and Lipschitz assumptions and then summing a geometric series,

$$\|h_t\| \leq \lambda \|h_{t-1}\| + L_x \|x_t\| \leq \sum_{i=0}^{t} \lambda^i L_x B_x \leq \frac{L_x B_x}{(1 - \lambda)}.$$

Now, consider the difference between hidden states at time step $t$. Unrolling the iterates $k$ steps and then using the previous display yields

$$\left\| h_t - h_t^k \right\| = \left\| \phi_w(h_{t-1}, x_t) - \phi_w(h_{t-1}^k, x_t) \right\| \leq \lambda \left\| h_{t-1} - h_{t-1}^k \right\| \leq \lambda^k \|h_{t-k}\| \leq \frac{\lambda^k L_x B_x}{(1 - \lambda)},$$

and solving for $k$ gives the result. $\qquad \square$

## B.1 PROOFS FROM SECTION 3.3

Before proceeding, we introduce notation for our smoothness assumption. We assume the map $\phi_w$ satisfies four smoothness conditions: for any reachable states $h, h'$, and any weights $w, w' \in \Theta$, there are some scalars $\beta_{ww}, \beta_{wh}, \beta_{hw}, \beta_{hh}$ such that

1. $\left\| \frac{\partial \phi_w(h, x)}{\partial w} - \frac{\partial \phi_{w'}(h, x)}{\partial w} \right\| \leq \beta_{ww} \|w - w'\|.$

2. $\left\| \frac{\partial \phi_w(h, x)}{\partial w} - \frac{\partial \phi_w(h', x)}{\partial w} \right\| \leq \beta_{wh} \|h - h'\|.$

3. $\left\| \frac{\partial \phi_w(h, x)}{\partial h} - \frac{\partial \phi_{w'}(h, x)}{\partial h} \right\| \leq \beta_{hw} \|w - w'\|.$

4. $\left\| \frac{\partial \phi_w(h, x)}{\partial h} - \frac{\partial \phi_w(h', x)}{\partial h} \right\| \leq \beta_{hh} \|h - h'\|.$

### B.1.1 GRADIENT DIFFERENCE DUE TO TRUNCATION IS NEGLIGIBLE

In the section, we argue the difference in gradient with respect to the weights between the recurrent and truncated models is $O(k\lambda^k)$. For sufficiently large $k$ (independent of the sequence length), the impact of truncation is therefore negligible. The proof leverages the "vanishing-gradient" phenomenon– the long-term components of the gradient of the full recurrent model quickly vanish. The remaining challenge is to show the short-term components of the gradient are similar for the full and recurrent models.

*Proof of Lemma 2.* The Jacobian of the loss with respect to the weights is

$$\frac{\partial p_T}{\partial w} = \frac{\partial p_T}{\partial h_T} \left( \sum_{t=0}^{T} \frac{\partial h_T}{\partial h_t} \frac{\partial h_t}{\partial w} \right),$$

where $\frac{\partial h_t}{\partial w}$ is the partial derivative of $h_t$ with respect to $w$, assuming $h_{t-1}$ is constant with respect to $w$. Expanding the expression for the gradient, we wish to bound

$$\left\| \nabla_w p_T(w) - \nabla_w p_T^k(w) \right\| = \left\| \sum_{t=1}^{T} \left( \frac{\partial h_T}{\partial h_t} \frac{\partial h_t}{\partial w} \right)^\top \nabla_{h_T} p_T - \sum_{t=T-k+1}^{T} \left( \frac{\partial h_T^k}{\partial h_t^k} \frac{\partial h_t^k}{\partial w} \right)^\top \nabla_{h_T^k} p_T^k \right\|$$

$$\leq \left\| \sum_{t=1}^{T-k} \left( \frac{\partial h_T}{\partial h_t} \frac{\partial h_t}{\partial w} \right)^\top \nabla_{h_T} p_T \right\|$$

$$+ \sum_{t=T-k+1}^{T} \left\| \left( \frac{\partial h_T}{\partial h_t} \frac{\partial h_t}{\partial w} \right)^\top \nabla_{h_T} p_T - \left( \frac{\partial h_T^k}{\partial h_t^k} \frac{\partial h_t^k}{\partial w} \right)^\top \nabla_{h_T^k} p_T \right\|.$$

The first term consists of the "long-term components" of the gradient for the recurrent model. The second term is the difference in the "short-term components" of the gradients between the recurrent and truncated models. We bound each of these terms separately.

For the first term, by the Lipschitz assumptions, $\|\nabla_{h_T} p_T\| \leq L_p$ and $\|\nabla_w h_t\| \leq L_w$. Since $\phi_w$ is $\lambda$-contractive, so $\left\|\frac{\partial h_t}{\partial h_{t-1}}\right\| \leq \lambda$. Using submultiplicavity of the spectral norm,

$$\left\|\frac{\partial p_T}{\partial h_T}\sum_{t=0}^{T-k}\frac{\partial p_T}{\partial h_t}\frac{\partial h_t}{\partial w}\right\| \leq \|\nabla_{h_T}p_T\|\sum_{t=0}^{T-k}\left\|\prod_{i=t}^{T}\frac{\partial h_i}{\partial h_{i-1}}\right\|\|\nabla_w h_t\| \leq L_p L_w\sum_{t=0}^{T-k}\lambda^{T-t} \leq \lambda^k\frac{L_p L_w}{(1-\lambda)}.$$

Focusing on the second term, by triangle inequality and smoothness,

$$\sum_{t=T-k+1}^{T}\left\|\left(\frac{\partial h_T}{\partial h_t}\frac{\partial h_t}{\partial w}\right)^{\top}\nabla_{h_T}p_T - \left(\frac{\partial h_T^k}{\partial h_t^k}\frac{\partial h_t^k}{\partial w}\right)^{\top}\nabla_{h_T^k}p_T\right\|$$

$$\leq \sum_{t=T-k+1}^{T}\left\|\nabla_{h_T}p_T - \nabla_{h_T^k}p_T^k\right\|\left\|\frac{\partial h_T^k}{\partial h_t^k}\frac{\partial h_t^k}{\partial w}\right\| + \|\nabla_{h_T}p_T\|\left\|\frac{\partial h_T}{\partial h_t}\frac{\partial h_t}{\partial w} - \frac{\partial h_T^k}{\partial h_t^k}\frac{\partial h_t^k}{\partial w}\right\|$$

$$\leq \sum_{t=T-k+1}^{T}\underbrace{\beta_p\left\|h_T - h_T^k\right\|\lambda^{T-t}L_w}_{(a)} + \underbrace{L_p\left\|\frac{\partial h_T}{\partial h_t}\frac{\partial h_t}{\partial w} - \frac{\partial h_T^k}{\partial h_t^k}\frac{\partial h_t^k}{\partial w}\right\|}_{(b)}.$$

Using Lemma 1 to upper bound (a),

$$\sum_{t=T-k}^{T}\beta_p\left\|h_T - h_T^k\right\|\lambda^{T-t}L_w \leq \sum_{t=T-k}^{T}\lambda^{T-t}\frac{\lambda^k\beta_p L_w L_x B_x}{(1-\lambda)} \leq \frac{\lambda^k\beta_p L_w L_x B_x}{(1-\lambda)^2}.$$

Using the triangle inequality, Lipschitz and smoothness, (b) is bounded by

$$\sum_{t=T-k+1}^{T}L_p\left\|\frac{\partial h_T}{\partial h_t}\frac{\partial h_t}{\partial w} - \frac{\partial h_T^k}{\partial h_t^k}\frac{\partial h_t^k}{\partial w}\right\|$$

$$\leq \sum_{t=T-k+1}^{T}L_p\left\|\frac{\partial h_T}{\partial h_t}\right\|\left\|\frac{\partial h_t}{\partial w} - \frac{\partial h_t^k}{\partial w}\right\| + L_p\left\|\frac{\partial h_t^k}{\partial w}\right\|\left\|\frac{\partial h_T}{\partial h_t} - \frac{\partial h_T^k}{\partial h_t^k}\right\|$$

$$\leq \sum_{t=T-k+1}^{T}L_p\lambda^{T-t}\beta_{wh}\left\|h_t - h_t^k\right\| + L_p L_w\left\|\frac{\partial h_T}{\partial h_t} - \frac{\partial h_T^k}{\partial h_t^k}\right\|$$

$$\leq k\lambda^k\frac{L_p\beta_{wh}L_x B_x}{(1-\lambda)} + \underbrace{L_p L_w\sum_{t=T-k+1}^{T}\left\|\frac{\partial h_T}{\partial h_t} - \frac{\partial h_T^k}{\partial h_t^k}\right\|}_{(c)},$$

where the last line used $\left\|h_t - h_t^k\right\| \leq \lambda^{t-(T-k)}\frac{L_x B_x}{(1-\lambda)}$ for $t \geq T - k$. It remains to bound (c), the difference of the hidden-to-hidden Jacobians. Peeling off one term at a time and applying triangle inequality, for any $t \geq T - k + 1$,

$$\left\|\frac{\partial h_T}{\partial h_t} - \frac{\partial h_T^k}{\partial h_t^k}\right\| \leq \left\|\frac{\partial h_T}{\partial h_{T-1}} - \frac{\partial h_T^k}{\partial h_{T-1}^k}\right\|\left\|\frac{\partial h_{T-1}}{\partial h_t}\right\| + \left\|\frac{\partial h_T^k}{\partial h_{T-1}^k}\right\|\left\|\frac{\partial h_{T-1}}{\partial h_t} - \frac{\partial h_{T-1}^k}{\partial h_t^k}\right\|$$

$$\leq \beta_{hh}\left\|h_{T-1} - h_{T-1}^k\right\|\lambda^{T-t-1} + \lambda\left\|\frac{\partial h_{T-1}}{\partial h_t} - \frac{\partial h_{T-1}^k}{\partial h_t^k}\right\|$$

$$\leq \sum_{i=t}^{T-1}\beta_{hh}\lambda^{T-t-1}\left\|h_i - h_i^k\right\|$$

$$\leq \lambda^k\frac{\beta_{hh}L_x B_x}{(1-\lambda)}\sum_{i=t}^{T-1}\lambda^{i-t}$$

$$\leq \lambda^k\frac{\beta_{hh}L_x B_x}{(1-\lambda)^2},$$

so (c) is bounded by $k\lambda^k\frac{L_p L_w\beta_{hh}L_x B_x}{(1-\lambda)^2}$. Ignoring Lipschitz and smoothness constants, we've shown the entire sum is $O\left(\frac{k\lambda^k}{(1-\lambda)^2}\right)$. $\qquad\square$

### B.1.2 STABLE RECURRENT MODELS ARE SMOOTH

In this section, we prove that the gradient map $\nabla_w p_T$ is Lipschitz. First, we show on the forward pass, the difference between hidden states $h_t(w)$ and $h_t'(w')$ obtained by running the model with weights $w$ and $w'$, respectively, is bounded in terms of $\|w - w'\|$. Using smoothness of $\phi$, the difference in gradients can be written in terms of $\|h_t(w) - h_t'(w')\|$, which in turn can be bounded in terms of $\|w - w'\|$. We repeatedly leverage this fact to conclude the total difference in gradients must be similarly bounded.

We first show small differences in weights don't significantly change the trajectory of the recurrent model.

**Lemma 5.** *For some $w, w'$, suppose $\phi_w, \phi_{w'}$ are $\lambda$-contractive and $L_w$ Lipschitz in $w$. Let $h_t(w), h_t(w')$ be the hidden state at time $t$ obtain from running the model with weights $w, w'$ on common inputs $\{x_t\}$. If $h_0(w) = h_0(w')$, then*

$$\|h_t(w) - h_t(w')\| \leq \frac{L_w \|w - w'\|}{(1 - \lambda)}.$$

*Proof.* By triangle inequality, followed by the Lipschitz and contractivity assumptions,

$$\|h_t(w) - h_t(w')\|$$
$$= \|\phi_w(h_{t-1}(w), x_t) - \phi_{w'}(h_{t-1}(w'), x_t)\|$$
$$\leq \|\phi_w(h_{t-1}(w), x_t) - \phi_{w'}(h_{t-1}(w), x_t)\| + \|\phi_{w'}(h_{t-1}(w), x_t) - \phi_{w'}(h_{t-1}(w'), x_t)\|$$
$$\leq L_w \|w - w'\| + \lambda \|h_{t-1}(w) - h_{t-1}(w')\|.$$

Iterating this argument and then using $h_0(w) = h_0(w')$, we obtain a geometric series in $\lambda$.

$$\|h_t(w) - h_t(w')\| \leq L_w \|w - w'\| + \lambda \|h_{t-1}(w) - h_{t-1}(w')\|$$
$$\leq \sum_{i=0}^{t} L_w \|w - w'\| \lambda^i$$
$$\leq \frac{L_w \|w - w'\|}{(1 - \lambda)}. \qquad \square$$

The proof of Lemma 3 is similar in structure to Lemma 2, and follows from repeatedly using smoothness of $\phi$ and Lemma 5.

*Proof of Lemma 3.* Let $h_t' = h_t(w')$. Expanding the gradients and using $\|h_t(w) - h_t(w')\| \leq \frac{L_w \|w - w'\|}{(1-\lambda)}$ from Lemma 5.

$$\|\nabla_w p_T(w) - \nabla_w p_T(w')\|$$
$$\leq \sum_{t=1}^{T} \left\| \left( \frac{\partial h_T}{\partial h_t} \frac{\partial h_t}{\partial w} \right)^\top \nabla_{h_T} p_T - \left( \frac{\partial h_T'}{\partial h_t'} \frac{\partial h_t'}{\partial w} \right)^\top \nabla_{h_T'} p_T \right\|$$
$$\leq \sum_{t=1}^{T} \left\| \nabla_{h_T} p_T - \nabla_{h_T'} p_T \right\| \left\| \frac{\partial h_T'}{\partial h_t'} \frac{\partial h_t'}{\partial w} \right\| + \|\nabla_{h_T} p_T\| \left\| \frac{\partial h_T}{\partial h_t} \frac{\partial h_t}{\partial w} - \frac{\partial h_T'}{\partial h_t'} \frac{\partial h_t'}{\partial w} \right\|$$
$$\leq \sum_{t=1}^{T} \beta_p \|h_T - h_T'\| \lambda^{T-t} L_w + L_p \left\| \frac{\partial h_T}{\partial h_t} \frac{\partial h_t}{\partial w} - \frac{\partial h_T'}{\partial h_t'} \frac{\partial h_t'}{\partial w} \right\|$$
$$\leq \frac{\beta_p L_w^2 \|w - w'\|}{(1 - \lambda)^2} + L_p \underbrace{\sum_{t=1}^{T} \left\| \frac{\partial h_T}{\partial h_t} \frac{\partial h_t}{\partial w} - \frac{\partial h_T'}{\partial h_t'} \frac{\partial h_t'}{\partial w} \right\|}_{(a)}.$$

Focusing on term (a),

$$L_p \sum_{t=1}^{T} \left\| \frac{\partial h_T}{\partial h_t} \frac{\partial h_t}{\partial w} - \frac{\partial h'_T}{\partial h'_t} \frac{\partial h'_t}{\partial w} \right\|$$

$$\leq L_p \sum_{t=1}^{T} \left\| \frac{\partial h_T}{\partial h_t} - \frac{\partial h'_T}{\partial h'_t} \right\| \left\| \frac{\partial h_t}{\partial w} \right\| + L_p \left\| \frac{\partial h'_T}{\partial h'_t} \right\| \left\| \frac{\partial h_t}{\partial w} - \frac{\partial h'_t}{\partial w} \right\|$$

$$\leq L_p L_w \sum_{t=1}^{T} \left\| \frac{\partial h_T}{\partial h_t} - \frac{\partial h'_T}{\partial h'_t} \right\| + L_p \sum_{t=1}^{T} \lambda^{T-t} \left( \beta_{wh} \|h_t - h'_t\| + \beta_{ww} \|w - w'\| \right)$$

$$\leq \underbrace{L_p L_w \sum_{t=1}^{T} \left\| \frac{\partial h_T}{\partial h_t} - \frac{\partial h'_T}{\partial h'_t} \right\|}_{(b)} + \frac{L_p \beta_{wh} L_w \|w - w'\|}{(1-\lambda)^2} + \frac{L_p \beta_{ww} \|w - w'\|}{(1-\lambda)},$$

where the penultimate line used,

$$\left\| \frac{\partial h_t}{\partial w} - \frac{\partial h'_t}{\partial w} \right\| \leq \left\| \frac{\partial \phi_w(h_{t-1}, x_t)}{\partial w} - \frac{\partial \phi_w(h'_{t-1}, x_t)}{\partial w} \right\| + \left\| \frac{\partial \phi_w(h'_{t-1}, x_t)}{\partial w} - \frac{\partial \phi_{w'}(h'_{t-1}, x_t)}{\partial w} \right\|$$

$$\leq \beta_{wh} \|h - h'\| + \beta_{ww} \|w - w'\|.$$

To bound (b), we peel off terms one by one using the triangle inequality,

$$L_p L_w \sum_{t=1}^{T} \left\| \frac{\partial h_T}{\partial h_t} - \frac{\partial h'_T}{\partial h'_t} \right\|$$

$$\leq L_p L_w \sum_{t=1}^{T} \left\| \frac{\partial h_T}{\partial h_{T-1}} - \frac{\partial h'_T}{\partial h'_{T-1}} \right\| \left\| \frac{\partial h_{T-1}}{\partial h_t} \right\| + \left\| \frac{\partial h'_T}{\partial h'_{T-1}} \right\| \left\| \frac{\partial h_{T-1}}{\partial h_t} - \frac{\partial h'_{T-1}}{\partial h'_t} \right\|$$

$$\leq L_p L_w \sum_{t=1}^{T} \left[ \left( \beta_{hh} \|h_{T-1} - h'_{T-1}\| + \beta_{hw} \|w - w'\| \right) \lambda^{T-t-1} + \lambda \left\| \frac{\partial h_{T-1}}{\partial h_t} - \frac{\partial h'_{T-1}}{\partial h'_t} \right\| \right]$$

$$\leq L_p L_w \sum_{t=1}^{T} \left[ \beta_{hw}(T-t)\lambda^{T-t-1} \|w - w'\| + \beta_{hh} \sum_{i=1}^{T-t} \|h_{T-i} - h'_{T-i}\| \lambda^{T-t-1} \right]$$

$$\leq L_p L_w \sum_{t=1}^{T} \left[ \beta_{hw}(T-t)\lambda^{T-t-1} \|w - w'\| + \frac{\beta_{hh} L_w \|w - w'\|}{(1-\lambda)} (T-t)\lambda^{T-t-1} \right]$$

$$\leq \frac{L_p L_w \beta_{hw} \|w - w'\|}{(1-\lambda)^2} + \frac{L_p L_w^2 \beta_{hh} \|w - w'\|}{(1-\lambda)^3}.$$

Supressing Lipschitz and smoothness constants, we've shown the entire sum is $O(1/(1-\lambda)^3)$, as required. $\qquad\square$

### B.1.3 GRADIENT DESCENT ANALYSIS

Equipped with the smoothness and truncation lemmas (Lemmas 2 and 3), we turn towards proving the main gradient descent result.

*Proof of Proposition 4.* Let $\Pi_\Theta$ denote the Euclidean projection onto $\Theta$, and let $\delta_i = \|w_{\text{recurr}}^i - w_{\text{trunc}}^i\|$. Initially $\delta_0 = 0$, and on step $i+1$, we have the following recurrence rela-

tion for $\delta_{i+1}$,

$$
\begin{aligned}
\delta_{i+1} &= \left\| w_{\text{recurr}}^{i+1} - w_{\text{trunc}}^{i+1} \right\| \\
&= \left\| \Pi_\Theta(w_{\text{recurr}}^i - \alpha_i \nabla p_T(w^i)) - \Pi_\Theta(w_{\text{trunc}}^i - \alpha_i \nabla p_T^k(w_{\text{trunc}}^i)) \right\| \\
&\le \left\| w_{\text{recurr}}^i - \alpha_i \nabla p_T(w^i)) - w_{\text{trunc}}^i - \alpha_i \nabla p_T^k(w_{\text{trunc}}^i) \right\| \\
&\le \left\| w_{\text{recurr}}^i - w_{\text{trunc}}^i \right\| + \alpha_i \left\| \nabla p_T(w_{\text{recurr}}^i) - \nabla p_T^k(w_{\text{trunc}}^i) \right\| \\
&\le \delta_i + \alpha_i \left\| \nabla p_T(w_{\text{recurr}}^i) - \nabla p_T(w_{\text{trunc}}^i) \right\| + \alpha_i \left\| \nabla p_T(w_{\text{trunc}}^i) - \nabla p_T^k(w_{\text{trunc}}^i) \right\| \\
&\le \delta_i + \alpha_i \left( \beta \delta_i + \gamma k \lambda^k \right) \\
&\le \exp\left( \alpha_i \beta \right) \delta_i + \alpha_i \gamma k \lambda^k,
\end{aligned}
$$

the penultimate line applied lemmas 2 and 3, and the last line used $1 + x \le e^x$ for all $x$. Unwinding the recurrence relation at step $N$,

$$
\begin{aligned}
\delta_N &\le \sum_{i=1}^N \left\{ \prod_{j=i+1}^N \exp(\alpha_j \beta) \right\} \alpha_i \gamma k \lambda^k \\
&\le \sum_{i=1}^N \left\{ \prod_{j=i+1}^N \exp\left( \frac{\alpha \beta}{j} \right) \right\} \frac{\alpha \gamma k \lambda^k}{i} \\
&= \sum_{i=1}^N \left\{ \exp\left( \alpha \beta \sum_{j=i+1}^N \frac{1}{j} \right) \right\} \frac{\alpha \gamma k \lambda^k}{i}.
\end{aligned}
$$

Bounding the inner summation via an integral, $\sum_{j=i+1}^N \frac{1}{j} \le \log(N/i)$ and simplifying the resulting expression,

$$
\begin{aligned}
\delta_N &\le \sum_{i=1}^N \exp(\alpha \beta \log(N/i)) \frac{\alpha \gamma k \lambda^k}{i} \\
&= \alpha \gamma k \lambda^k N^{\alpha\beta} \sum_{i=1}^N \frac{1}{i^{\alpha\beta+1}} \\
&\le \alpha \gamma k \lambda^k N^{\alpha\beta+1}.
\end{aligned}
$$

$\square$

### B.1.4 PROOF OF THEOREM 1

*Proof of Theorem 1.* Using $f$ is $L_f$-Lipschitz and the triangle inequality,

$$
\begin{aligned}
\left\| y_T - y_T^k \right\| &\le L_f \left\| h_T(w_{\text{recurr}}^N) - h_T^k(w_{\text{trunc}}^N) \right\| \\
&\le L_f \left\| h_T(w_{\text{recurr}}^N) - h_T(w_{\text{trunc}}^N) \right\| + L_f \left\| h_T(w_{\text{trunc}}^N) - h_T^k(w_{\text{trunc}}^N) \right\|.
\end{aligned}
$$

By Lemma 5, the first term is bounded by $\frac{L_w \left\| w_{\text{recurr}}^N - w_{\text{trunc}}^N \right\|}{(1-\lambda)}$, and by Lemma 1, the second term is bounded by $\lambda^k \frac{L_x B_x}{(1-\lambda)}$. Using Proposition 4, after $N$ steps of gradient descent, we have

$$
\begin{aligned}
\left\| y_T - y_T^k \right\| &\le \frac{L_f L_w \left\| w_{\text{recurr}}^N - w_{\text{trunc}}^N \right\|}{(1-\lambda)} + \lambda^k \frac{L_f L_x B_x}{(1-\lambda)} \\
&\le k \lambda^k \frac{\alpha L_f L_w N^{\alpha\beta+1}}{(1-\lambda)} + \lambda^k \frac{L_f L_x B_x}{(1-\lambda)},
\end{aligned}
$$

and solving for $k$ such that both terms are less than $\varepsilon/2$ gives the result. $\square$

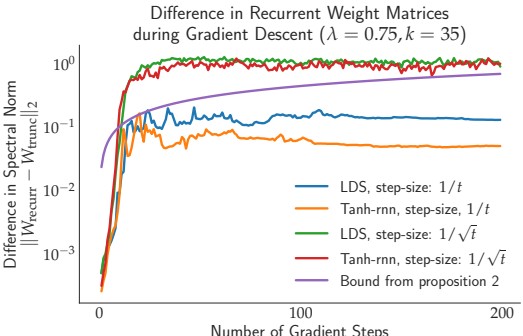

Figure 6: Empirical validation Proposition 4 on random Gaussian instances. Without the $1/t$ rate, the gradient descent bound no longer appears qualitatively correct, suggesting the $O(1/t)$ rate is necessary.

## C EXPERIMENTS

**The $O(1/t)$ rate may be necessary.** The key result underlying Theorem 1 is the bound on the parameter difference $\|w_{\text{trunc}} - w_{\text{recurr}}\|$ while running gradient descent obtained in Proposition 4. We show this bound has the correct qualitative scaling using random instances and training randomly initialized, stable linear dynamical systems and $\tanh$-RNNs. In Figure 6, we plot the parameter error $\|w_{\text{trunc}}^t - w_{\text{recurr}}^t\|$ as training progresses for both models (averaged over 10 runs). The error scales comparably with the bound given in Proposition 4. We also find for larger step-sizes like $\alpha/\sqrt{t}$ or constant $\alpha$, the bound fails to hold, suggesting the $O(1/t)$ condition is necessary.

Concretely, we generate random problem instance by fixing a sequence length $T = 200$, sampling input data $x_t \overset{\text{i.i.d.}}{\sim} \mathcal{N}(0, 4 \cdot I_{32})$, and sampling $y_T \sim \text{Unif}[-2, 2]$. Next, we set $\lambda = 0.75$ and randomly initialize a stable linear dynamical system or RNN with $\tanh$ non-linearity by sampling $U_{ij}, W_{ij} \overset{\text{i.i.d.}}{\sim} \mathcal{N}(0, 0.5)$ and thresholding the singular values of $W$ so $\|W\| \leq \lambda$. We use the squared loss and prediction function $f(h_t, x_t) = Ch_t + Dx_t$, where $C, D \overset{\text{i.i.d.}}{\sim} \mathcal{N}(0, I_{32})$. We fix the truncation length to $k = 35$, set the learning rate to $\alpha_t = \alpha/t$ for $\alpha = 0.01$, and take $N = 200$ gradient steps. These parameters are chosen so that the $\gamma k \lambda^k N^{\alpha\beta+1}$ bound from Proposition 4 does not become vacuous – by triangle inequality, we always have $\|w_{\text{trunc}} - w_{\text{recurr}}\| \leq 2\lambda$.

**Stable vs. unstable models.** The word and character level language modeling experiments are based on publically available code from Merity et al. (2018). The polyphonic music modeling code is based on the code in Bai et al. (2018), and the slot-filling model is a reimplementation of Mesnil et al. (2015) [1]

Since the sufficient conditions for stability derived in Section 2.2 only apply for networks with a single layer, we use a single layer RNN or LSTM for all experiments. Further, our theoretical results are only applicable for vanilla SGD, and not adaptive gradient methods, so all models are trained with SGD. Table 2 contains a summary of all the hyperparameters for each experiment.

All hyperparameters are shared between the stable and unstable variants of both models. In the RNN case, enforcing stability is conceptually simple, though computationally expensive. Since $\tanh$ is 1-Lipschitz, the RNN is stable as long as $\|W\| < 1$. Therefore, after each gradient update, we project $W$ onto the spectral norm ball by taking the SVD and thresholding the singular values to lie in $[0, 1)$. In the LSTM case, enforcing stability is conceptually more difficult, but computationally simple. To ensure the LSTM is stable, we appeal to Proposition 2. We enforce the following inequalities after each gradient update

---

[1]The word-level language modeling code is based on `https://github.com/pytorch/examples/tree/master/word_language_model`, the character-level code is based on `https://github.com/salesforce/awd-lstm-lm`, and the polyphonic music modeling code is based on `https://github.com/locuslab/TCN`.

Table 2: Hyperparameters for all experiments

|  |  | Model | |
| --- | --- | --- | --- |
|  |  | RNN | LSTM |
| **Word LM** | Number layers | 1 | 1 |
|  | Hidden units | 256 | 1024 |
|  | Embedding size | 1024 | 512 |
|  | Dropout | 0.25 | 0.65 |
|  | Batch size | 20 | 20 |
|  | Learning rate | 2.0 | 20. |
|  | BPTT | 35 | 35 |
|  | Gradient clipping | 0.25 | 1.0 |
|  | Epochs | 40 | 40 |
| **Char LM** | Number layers | 1 | 1 |
|  | Hidden units | 768 | 1024 |
|  | Embedding size | 400 | 400 |
|  | Dropout | 0.1 | 0.1 |
|  | Weight decay | 1e-6 | 1e-6 |
|  | Batch size | 80 | 80 |
|  | Learning rate | 2.0 | 20.0 |
|  | BPTT | 150 | 150 |
|  | Gradient clipping | 1.0 | 1.0 |
|  | Epochs | 300 | 300 |
| **Polyphonic Music** | Number layers | 1 | 1 |
|  | Hidden units | 1024 | 1024 |
|  | Dropout | 0.1 | 0.1 |
|  | Batch size | 1 | 1 |
|  | Learning rate | 0.05 | 2.0 |
|  | Gradient clipping | 5.0 | 5.0 |
|  | Epochs | 100 | 100 |
| **Slot-Filling** | Number layers | 1 | 1 |
|  | Hidden units | 128 | 128 |
|  | Embedding size | 64 | 64 |
|  | Dropout | 0.5 | 0.5 |
|  | Weight decay | 1e-4 | 1e-4 |
|  | Batch size | 128 | 128 |
|  | Learning rate | 10.0 | 10.0 |
|  | Gradient clipping | 1.0 | 1.0 |
|  | Epochs | 100 | 100 |

1. The hidden-to-hidden forget gate matrix should satisfy $\|W_f\|_\infty < 0.128$, which is enforced by normalizing the $\ell_1$- norm of each row to have value at most $0.128$.

2. The input vectors $x_t$ must satisfy $\|x_t\|_\infty \leq B_x = 0.75$, which is achieved by thresholding all values to lie in $[-0.75, 0.75]$.

3. The bias of the forget gate $b_f$, must satsify $\|b_f\|_\infty \leq 0.25$, which is again achieved by thresholding all values to lie in $[-0.25, 0.25]$.

4. The input-hidden forget gate matrix $U_f$ should satisfy $\|U_f\|_\infty \leq 0.25$. This is enforced by normalizing the $\ell_1$- norm of each row to have value at most $0.25$.

5. Given 1-4, the forget gate can take value at most $f_\infty < 0.64$. Consequently, we enforce $\|W_i\|_\infty, \|W_o\|_\infty \leq 0.36$, $\|W_z\| \leq 0.091$, and $\|W_f\|_\infty < \min\{0.128, (1-0.64)^2\} = 0.128$.

After 1-5 are enforced, by Proposition 2, the resulting (iterated)-LSTM is stable. Although the above description is somewhat complicated, the implementation boils down to normalizing the rows of the LSTM weight matrices, which can be done very efficiently in a few lines of PyTorch.

**Data-dependent stability.** Unlike the RNN, in an LSTM, it is not clear how to analytically compute the stability parameter $\lambda$. Instead, we rely on a heuristic method to estimate $\lambda$. Recall a model is stable if for all $x, h, h'$, we have

$$S(h, h', x) := \frac{\|\phi_w(h, x) - \phi_w(h', x)\|}{\|h - h'\|} \leq \lambda < 1. \tag{7}$$

To estimate $\sup_{h,h',x} S(h, h', x)$, we do the following. First, we take $x$ to be point in the training set. In the language modeling case, $x$ is one of the learned word-vectors. We randomly sample and fix $x$, and then we perform gradient ascent on $S(h, h', x)$ to find worst-case $h, h'$. In our experiments, we initialize $h, h' \sim \mathcal{N}(0, 0.1 \cdot I)$ and run gradient ascent with learning rate $0.9$ for $1000$ steps. This procedure is repeated 20 times, and we estimate $\lambda$ as the maximum value of $S(h, h', x)$ encounted during any iteration from any of the 20 random starting points.

