# OpenReview forum: "Stable Recurrent Models"
_ICLR.cc/2019/Conference_

### Official Review · AnonReviewer3 · 2018-11-01

**Rating:** 6
**Confidence:** 4

**Review:**

In this paper, the authors study the stability property of recurrent neural networks. Adopting the definition of stability from the dynamical system literature, the authors present a generic definition of stable recurrent models and provide sufficient conditions of stable linear RNNs and LSTMs. The authors also study the "feed-forward" approximation of recurrent networks and theoretically show that the approximation works for both inference and training. Experimental studies compare the performance of stable and unstable models on various tasks.

The paper is well-written and very pleasant to read. The notations are clear and the claims are relatively easy to follow. The theoretical analysis in Section 3 is novel, interesting and solid. However, the reviewer has concerns about the motivation of the presented analysis and insufficient empirical results.

The stability property only eliminates the exploding gradient problem, but not the vanishing gradient problem. The reviewer suspects that a stable recurrent model always suffers from vanishing gradient. Therefore, stability might not necessarily be a desirable property. There has been a line of work that constrain the weight matrix in RNNs to be orthogonal or unitary so that the gradient won't explode, e.g. [1], [2], [3]. It seems that the orthogonal or unitary conditions are stronger than the stability condition, and are probably less prone to the vanishing gradient problem.

The vanishing gradient problem is also related to the analysis in Section 3. If a recurrent network is very stable and has vanishing gradient, then a small perturbation of the initial hidden state has little effect on later time steps. This intuitively explains why it can be well approximated by using only the last k time steps. However, the recurrent model itself might not be a desirable model.  In other words, although Theorem 1 shows that $y_T$ and $y_T^k$ can be arbitrarily close, $y_T$ might not be a good prediction.

The experimental study seems weak. Again, in the RNN case, constraining the singular values of the weight matrix is not a new idea. Furthermore, the results in Table 1 seem to suggest that the stable models perform worse than unstable ones. What is the benefit in using stable models? Proposition 2 is only a sufficient condition of a stable LSTM and it seems very restrictive, as the authors point out. This might explain the worse performance of the stable LSTMs in Table 1. The reviewer was expecting more experimental results to support the claims in Section 3. For example, an empirically study of the difference between a recurrent model and a "feed-forward" or truncation approximation.

Minor comments:
* Lemma 1: $\lambda$-contractive => $\lambda$-contractive in $h$?
* Theorem 1: $k=O(...)$ => $k=\Omega(...)$? Intuitively, a bigger k leads to a better feed-forward approximation.

[1] Martin Arjovsky, Amar Shah, and Yoshua Bengio. Unitary evolution recurrent neural networks. ICML, 2016.
[2] Scott Wisdom, Thomas Powers, John Hershey, Jonathan Le Roux, and Les Atlas. Full-capacity unitary recurrent neural networks. NIPS, 2016.
[3] Eugene Vorontsov, Chiheb Trabelsi, Samuel Kadoury, and Chris Pal. On orthogonality and learning recurrent networks with long term dependencies. ICML, 2017.

---

> ### Author Response · Authors · 2018-11-10
> **Response to Reviewer 3**
>
> Thank you for your detailed comments and feedback. We have incorporated some of these suggestions into a revision of the paper. We discuss your concerns below.
>
> Motivation of stable models:
> There are two reasons to consider stability in recurrent models:
> 1) Stability is natural criterion for learnability in recurrent models. Outside the stable regime, learning recurrent models requires a delicate mix of heuristics. Studying stable models addresses whether this collection of tricks is actually necessary, and our results suggest a better-behaved model class can solve many of the same problems.
>
> 2) Understanding whether models trained in practice are in the stable regime helps answer when recurrent models are truly necessary. As the reviewer noted, whether the stable model is “desirable” depends on experimentation. However, when a stable model achieves similar performance with an unstable model, the conclusion is a feed-forward network suffices to solve the task. We demonstrate sequence learning happens in the stable regime, and this helps explain the widespread success of feed-forward models on sequence problems.
>
>
> Vanishing Gradients:
> Stable recurrent models always have vanishing gradients, and vanishing gradients are an important part of proving our approximation results. However, vanishing gradients are not unique to stable models. In the updated version of the paper, we show unstable language models also exhibit vanishing gradients. This corroborates the evidence in section 4.3 showing these models operate in the stable regime.
>
> The cited unitary RNN models may help reduce vanishing gradients. Even in these works, there is still gradient decay over time (e.g. Figure 4, ii in [1]), but the rate of decay is slower. The updated version of the paper includes a brief discussion of these works. At minimum, these models have not yet seen widespread use, and our work demonstrates models frequently trained in practice are either stable or can be made stable without performance loss.
>
> Empirical study of the difference between recurrent and truncated models:
> In the revision, we added experiments studying truncation in the unstable models and also show unstable models satisfy a qualitative version of Theorem 1. All of the models considered, including the LSTM language models, exhibit sharply diminishing returns to larger values of the truncation parameter. As predicted by theorem 1, the difference between the truncated and full recurrent matrix during training becomes small for moderate values of the truncation parameter.
>
> Comparison between stable and unstable models:
> We disagree with the interpretation of Table 1. Except for the LSTM language models, the variation in performance between stable and unstable models is within standard-error. We do not retune the hyperparameters when imposing stability, and the near equivalence of the results is evidence the unstable models do not offer a large performance boost. For the LSTM language models, in section 4.3 and 4.4, we argue the unstable LSTM language models are close to the stable regime, and the gap between stable and unstable models is an artifact of the particular way we impose stability.

---

> > ### Comment · AnonReviewer3 · 2018-11-28
> > **Thanks for the response**
> >
> > Thanks for the clarification and fixing the notations in Theorem 1. I think the discussion of unitary RNN models makes the paper more well-rounded. I hope this work will inspire more research in this direction in the future and help us understand the dynamics of recurrent networks. I would like to keep my rating.

---

### Official Review · AnonReviewer2 · 2018-11-05
**Interesting theoretical and practical results but false claims on RNNs**

**Rating:** 6
**Confidence:** 4

**Review:**

+ An interesting problem to study on the stability of RNNs
+ Investigation of spectral normalization to sequential predictions is worthwhile, especially Figure 2
+ Some theoretical justification of SGD for learning dynamic systems following Hardt et al. (2016b).

- The take-home message of the paper is not clear. First, it defines a  notion of stability based on Lipchitz-continuity and proves SGD can learn it. Then the experiments show such a definition is actually not correct, but rather a data-dependent one.
- The theory only looks at the instantaneous dynamics from time t to t+1, without unrolling the RNNs over time. Then it is not much different from analyzing feed-forward networks. The theorem on SGD is remotely related to the contribution of the paper.
- The spectral normalization technique that is actually used in experiments is not new

---

> ### Author Response · Authors · 2018-11-10
> **Response to reviewer 2**
>
> Thank you for your comments and feedback. We address each of your concerns below.
>
> Take-home message:
> The message of the paper is that sequence learning happens, or can be made to happen, in the stable regime. The Lipschitz definition of stability (eq. 2) and the “data-dependent” definition introduced in the experiments are complementary. The data-dependent definition is just a relaxation of the Lipschitz criteria-- we only require equation 2 to hold for inputs from the data-distribution. For the proofs and the majority of the experiments, the strict Lipschitz condition suffices. Most models can be made stable in the sense of equation 2 without performance loss. For LSTMs on language modeling, the data-dependent version illustrates even the nominally unstable LSTMs are close to the stable regime-- a truly unstable model would not satisfy even this weaker definition. We view results with both definitions as evidence recurrent models trained in practice operate in the stable regime.
>
> Instantaneous dynamics:
> The theory in our paper does consider unrolling the RNNs over time.  While the stability condition is stated purely in terms of the the state-transition function from step t to step t+1, the main theoretical results (Proposition 3 and Theorem 1) specifically concern the unrolled RNN. In particular, our results show that the unrolled (stable) RNN can be approximated by a feed-forward network.
>
> Spectral Normalization:
> In our experiments, our focus is more on comparing the performance of stable and unstable models and less on the particular form of normalization used to achieve stability. In the RNN case, enforcing stability via constraining the spectral norm of the recurrent matrix is fairly routine. In the LSTM case, the stability conditions given in Proposition 2 are new and allow one to experiment with stable LSTMs. The updated version of the paper includes a discussion of these other works.

---

> > ### Comment · AnonReviewer2 · 2018-11-12
> > **significance of the theoretical claim**
> >
> > - there is a gap between 'Lipschitz' and 'data-dependent' stability. why is that? In the proof of Section 2.2,  in order to satisfy the contractive mapping condition,  input data x does not have subscript t, can you justify?
> >
> > - the global stability property for one-layer RNN based on the Lipschitz condition of the activation function is a known result (e.g.[1]). what is the new contribution here?
> >
> > Jin, Liang, Peter N. Nikiforuk, and Madan M. Gupta. "Absolute stability conditions for discrete-time recurrent neural networks." IEEE Transactions on Neural Networks 5.6 (1994): 954-964.
> >
> > - The equivalence between RNN and feedforward networks is at the equilibrium state. But how about non-equilibrium states? and the number of weights? It is misleading to claim the two to be equivalent.

---

> > > ### Author Response · Authors · 2018-11-12
> > > **Response to Reviewer 2**
> > >
> > > Thank you for your prompt response. We address these concerns in turn.
> > >
> > > Gap between stability conditions:
> > > The data-dependent condition is a strict relaxation of the Lipschitz condition. Two additional comments are in order.
> > > 1) Stability is a clarifying concept. The Lipschitz condition is clean and allows us to understand the core phenomena associated with stability. The data-dependent definition is a useful diagnostic--  when our sufficient (Lipschitz) stability conditions fail to hold, the data-dependent condition addresses whether the model is still operating in the stable regime.
> > > 2) In many cases, we can still prove results with the data-dependent guarantee.
> > > --If the input representation is fixed, then all of the proofs go through with the data-dependent condition. If S is the set of inputs from the data distribution, we can simply replace all instances of “for all x” with “for all x in S”. This is the case with polyphonic music modeling.
> > > --When the input is not fixed (e.g. word vectors that are updated during training), the proofs go through provided S is interpreted as “all word vectors generated during training.”
> > >
> > > In section 2.2, the subscript t is dropped because the Lipschitz definition of stability (eq 2) must hold for all x.
> > >
> > > Theoretical contribution:
> > > Our main theoretical contribution is feed-forward approximation of stable recurrent models, especially Proposition 3 and Theorem 1. The results in section 2.2 give concrete examples of our general stability definition. For a 1-layer RNN, the cited paper [1] gives similar stability conditions. However, [1] does not touch on the question of feed-forward approximation, particularly approximation during training, nor does it mention LSTMs. We will add the appropriate citation, but note the RNN stability conditions are a routine one-line calculation and far from our main technical contribution.
> > >
> > > Equilibrium states:
> > > We only claim equivalence between *stable* RNNs and feed-forward networks. In stable RNNs, all trajectories converge to an equilibrium state. Certainly, general (unstable) RNNs cannot be approximated with feed-forward networks. Understanding to what extent models trained in practice are stable or can be made stable is then an empirical question, and we address this question in Section 4.
> > >
> > > Implementing truncated models as feed-forward networks increases the number of weights by a factor of $k$. This increase is an artifact of our analysis, and it is an interesting open question to find more parsimonious approximations. From a memory perspective, a feed-forward network with more weights is still a feed-forward network, and our result establishes stable recurrent models cannot have more memory than feed-forward models.

---

> > > > ### Comment · AnonReviewer2 · 2018-11-12
> > > > **Reasonable response**
> > > >
> > > > Appreciate your response.  I am willing to upgrade the rating if the authors can tone down the theoretical claims.

---

> > > > > ### Author Response · Authors · 2018-11-14
> > > > > **Revision to paper**
> > > > >
> > > > > Thank you for your response. We have updated the paper to reflect our discussion. In particular,
> > > > > - we make clear the sufficient stability conditions are only new in the case of the LSTM and appropriately cite Jin et al. for the 1-layer RNN
> > > > > - we added a discussion around the relationship between stability and data-dependent stability
> > > > > - we clarify our notion of "equivalence" is only in terms of the context required to make predictions and not, e.g., in terms of number of parameters or some other measure, and added further discussion of this distinction to Section 5.
> > > > >
> > > > > We're happy to address any additional concerns with the current presentation.

---

### Official Review · AnonReviewer1 · 2018-11-08
**Interesting theoretical angle on RNNs that provides insights but also feel incomplete**

**Rating:** 7
**Confidence:** 2

**Review:**

This is an interesting paper that I expect will generate some interest within the ICLR community and from deep learning researchers in general. The definition of stability is both intuitive and sound and the connection to exploding gradients is perhaps the most interesting and useful part of the paper. The sufficient conditions yield practical techniques for increasing the stability of, e.g., an LSTM, by constraining the weight matrices. They also show that stable recurrent models can be approximated by models with finite historical windows, e.g., truncated RNNs. Experiments in Sec 4 suggest that stable models produced by constraining standard RNN architectures can compete with their unconstrained unstable counterparts, and often without necessitating significant changes to architecture or hyperparameters. The perhaps most interesting observations are in Sec 4.3, in which the authors claim that even fundamentally unstable models, e.g., unconstrained RNNs, often operate in a stable regime, at least when being applied to in-sample data. I lean toward acceptance at the moment, but I am eager to discuss with the authors and other reviewers as I am not 100% confident that I fully understood the theory.

SUMMARY

This paper proposes a simple, generic definition of “stability” for recurrent, non-linear dynamical systems such as RNNs: that given two hidden states h, h’, the difference between their updated states given input x is bounded by the product between the difference between the states themselves and a small multiplier. The paper then immediately draws a connection between stability, asserting that unstable models are prone to gradient explosions during gradient descent-based training. In Sec 2.2, the paper presents sufficient conditions for basic RNNs and LSTMs to be stable. Secs 3.2 and 3.3 argue that stable recurrent models can be approximated by feedforward models during both inference and training with a finite history horizon, such as a RNN with a truncated history. Experiments in language and music modeling substantiate this claim: constrained, stable models are competitive with standard unconstrained models. Sec 4.3 sheds some light on this phenomenon, arguing that there is a weaker form of data-dependent stability and that even unstable models may operate in a stable regime for some problems, thus explaining the parity between stable and unstable models.

STRENGTHS

* This paper is surprisingly engaging and easy to read.
* The theorems are clearly stated and the proofs appear sound to me, though I will admit that I am not confident that I would catch a significant bug.
* This paper provides a new (to me, anyway) and thought-provoking analysis of RNNs. In particular, I was especially interested in the observation that stable models can be approximated by truncated models and that there is a connection between stability and long-term dependencies. This seems consistent with the fact that for many problems, non-recurrent models (ConvNets, Transformers, etc.) are often competitive with more complex architectures.

WEAKNESSES

* In practice it seems as though stability may depend on not only choice of  model architecture but also the data themselves. There is probably no good way to know a priori what the stability characteristics of a given data set are, making it tough to apply the ideas of this paper in practice
* The literature review seems a bit limited and appears to ignore the growing body of work on constraining RNN weight matrices to address both exploding and vanishing gradients. For example, I am pretty confident that the singular thresholding trick for renormalizing neural net weights has been  described in the literature previously.
* Although stable and unstable models appear to be competitive in experiments, the theoretical analysis provides no insights into stability and how it relates to accuracy.

---

> ### Author Response · Authors · 2018-11-10
> **Response to Reviewer 1**
>
> Thank you for your detailed comments and feedback.
>
> We agree it is difficult to know a priori whether particular dataset will be amenable to stable models. However, stability can still be a clarifying idea in practice. Given a dataset where stable models perform comparably with unstable models, either the dataset does not require long-term memory (i.e. feed-forward approximation suffices), or the unstable models do not take advantage of it. We conjecture most recurrent models successfully trained in practice are operating in the stable regime. To further test this claim, it would be interesting to find datasets (if any) where unstable models significantly outperform stable models, or datasets where non-recurrent models aren’t competitive with their recurrent counterparts.
>
> In the revision, we added discussion of the several recent works constraining RNN matrices. These works try to keep the model just outside the stable regime to avoid vanishing gradients and side-step exploding gradients (i.e. take lambda ~ 1). The spectral norm thresholding technique for RNNs is straightforward, whereas the stability conditions for the LSTM is new. In either case, our focus is on using these techniques to understand the consequences of imposing stability on recurrent models.
>
> In general, answering the question of accuracy is fairly delicate. We’re able to show stable and truncated/feed-forward models have the same accuracy. Bounds relating the accuracy of an unstable model with the accuracy of an stable one almost certainly require further assumptions on the data distribution. Obtaining such accuracy bounds for neural networks has been elusive, and part of the contribution of our work is proving a connection between the performance two model classes (stable RNNs and truncated/feed-forward models) without needing to resolve these questions.

---

> > ### Comment · AnonReviewer1 · 2018-11-10
> > **Thanks!**
> >
> > Thank you for the prompt and thoughtful response. I wanted to let you know that I have read it (and your other responses) and am thinking about follow-up questions. Expect me to reply by mid-next week.

---

> > > ### Comment · AnonReviewer1 · 2018-11-27
> > > **Well-written, thorough responses**
> > >
> > > I don't have much to add to the thorough discussion below. I was already in the "accept" camp, and I remain there. I will confer with the other reviewers and consider a revised score.

---

### Public Comment · (anonymous) · 2019-01-01
**Interesting take on RNN stability and needs small updates in Related Work**

I like the way the problem of RNN stability is tackled and the feasibility of replacing them with feed-forward networks is demonstrated in this paper.

There are a couple of papers on stabilizing RNN training which were published in ICML 2018 and NeurIPS 2018 which can be included into related work.

1) Stabilizing Gradients for Deep Neural Networks via Efficient SVD Parameterization - (Zhang et al, ICML 2018)
2) Kronecker Recurrent Units - (Jose et al, ICML 2018)
3) FastGRNN: A Fast, Accurate, Stable and Tiny Kilobyte Sized Gated Recurrent Neural Network - (Kusupati et al, NeurIPS 2018)

Also, in ICML 2017, along with (Vorontsov et al.) there were two more paper dealing with stabilization of RNNs
1)  Efficient orthogonal parametrisation of recurrent neural networks using householder reflections - (Mhammedi et al., ICML 2017)
2) Tunable Efficient Unitary Neural Networks (EUNN) and their application to RNNs - (Jing et al., ICML 2017).

It would be great if the authors could add these to the camera ready version of the paper make their related search more comprehensive and complete.

Thanks.

---

> ### Author Response · Authors · 2019-01-05
> **Thank you**
>
> Thank you for your interest in our paper and highlighting additional related work. We will incorporate these references into the final version of the paper.

---

### Meta-Review · Area_Chair1 · 2018-12-14
**Important topic, favorable reviews but are the stated implications general?**

**Confidence:** 5
**Recommendation:** Accept (Poster)

**Metareview:**

The paper presents both theoretical analysis (based upon lambda-stability) and experimental evidence on stability of recurrent neural networks. The results are convincing but is concerns with a restricted definition of stability. Even with this restriction acceptance is recommended.